# The Predictability of the Surgical Outcomes of Class III Patients in the Transverse Dimension—A Study of Three-Dimensional Assessment

**DOI:** 10.3390/jpm12071147

**Published:** 2022-07-15

**Authors:** U-Kei Lai, Cheng-Chun Wu, Yu-Jen Chang, Shiu-Shiung Lin, Jui-Pin Lai, Te-Ju Wu

**Affiliations:** 1Department of Craniofacial Orthodontics, Kaohsiung Chang Gung Memorial Hospital and Chang Gung University College of Medicine, 123 Dapi Rd. Niaosong District, Kaohsiung 833401, Taiwan; st951527@gmail.com (U.-K.L.); orthodontist.chang@gmail.com (Y.-J.C.); glasgow1993@yahoo.com (S.-S.L.); 2Department of Plastic Surgery, Kaohsiung Chang Gung Memorial Hospital and Chang Gung University College of Medicine, 123 Dapi Rd. Niaosong District, Kaohsiung 833401, Taiwan; bighelmet6@hotmail.com (C.-C.W.); benjplai@yahoo.com (J.-P.L.)

**Keywords:** orthognathic surgery, prognathism, malocclusion, Angle Class III

## Abstract

This study aimed to assess the outcomes of planned maxillary surgical movements in the transverse direction in patients possessing a Class III skeletal pattern. The available consecutive patients’ records were retrospectively reviewed. Only those possessing a Class III skeletal pattern, and for whom the same virtual planning system was used, were enrolled. The waferless technique was used to guide the jawbone repositioning. A representative triangle in the virtual maxilla of each stage was used to validate the planned surgical movements (PSMs) and the outcome discrepancy (OD). The linear and angular measurements were retrieved for the assessments of the correlation between PSM and OD. In total, 44 adult patients who met the inclusion criteria were studied. The average linear OD of the A-point in the transverse direction was 0.66 ± 0.54 mm, and the yaw correction showed 1.02 ± 0.84 degrees in difference. There was no specific correlation between the linear PSMs and ODs; however, the angular ones were positively correlated. With the help of the waferless technique to transfer the virtual planning results, the practitioners could confidently predict the postsurgical maxillary position in the transverse direction in the orthognathic surgery of Class III patients. However, the yaw correction should be carefully planned to avoid postsurgical instabilities.

## 1. Introduction

Orthognathic surgery (OGS) is a technically sensitive treatment that requires delicate interdisciplinary work to achieve the treatment goals and meet patients’ expectations. Computer-assisted surgical simulation has turned a new page in modern presurgical preparation in OGS [1]. With virtual planning, practitioners can easily and alternatively assess the feasibility of each possible plan, and the patient receives a more understandable illustration of the possible outcomes. All of these evolutions are helpful in the mutual communication between the patient and the medical teams.

Although presurgical simulation provides the vivid illustration of surgical results, the success of surgical intervention is largely dependent on the reliability of the guidance system for accurate intraoperative jawbone repositioning. Traditionally, an intermediate stent (wafer) placed over the unsplit mandible serves as the guide for the planned maxillary repositioning in OGS. However, the laboratory processing may cause unexpected fabrication errors [2], and it takes additional surgical time to fix and secure the stent during the guidance procedures. With the help of the three-dimensional printing technique, the evolving computer-assisted surgical simulation can provide a creational surgical guide to reposition the maxilla in the presurgical simulation without the intermediate stents. Such procedures use the so-called waferless technique [3,4,5,6]. The waferless technique using prebent or customized plates is one of the most popular guidance systems, reporting minimal repositioning errors [3,4,5].

The promising accuracy of jawbone movements implies not only the expectation of immediate surgical results, but also the chance to discuss the postsurgical changes over time. With an image-guided visualization display, Zinser et al. reported clinically acceptable precision six months after OGS via the waferless technique [6]. However, the scatter artifacts caused by brackets and metallic restoration, along with the smaller sample size, raised concern about the measurements. Recently, another study using a similar concept for jawbone repositioning reported stable maxillary position after 1-year follow-up. However, enrolling patients of different skeletal patterns in the limited samples weakened the findings [7].

Taking advantage of comprehensive three-dimensional investigation, the virtual system can provide valuable information about the postsurgical changes in the transverse dimension, which has rarely been discussed in previous cephalometric assessments because of their essential limitations. Therefore, this study aimed to assess the predictability of planned maxillary surgical movements in the transverse direction in patients with the same skeletal pattern and increased sample power. Furthermore, the possible confounding factors that affect the predictability of surgical outcomes in the transverse dimension were also investigated.

## 2. Materials and Methods

In the present study, the enrolled samples were retrieved from a database encompassing the medical records of patients receiving orthognathic surgery in the period from 2013 to 2017 at Kaohsiung Chang Gung Memorial Hospital, Kaohsiung, Taiwan. Only patients possessing a Class III skeletal pattern were included. All of the surgical plans were simulated by the same virtual system. Once the plans received the acceptance and confirmation of the patients, the surgical guidance appliances were fabricated with the help of the CAD/CAM technique. All of the enrolled patients received LeFort I without segmentation and bilateral sagittal split osteotomy (BSSO) in the maxilla-first approach performed by the same senior surgeon (J.P. Lai). On the other hand, patients with craniofacial disorders, facial trauma, or modified intraoperative treatment plans were excluded. The study design and data processing were approved by the Institutional Review Board of Kaohsiung Chang Gung Memorial Hospital (approval no. 201701645B0).

### 2.1. Presurgical Virtual Planning

For the virtual planning, all of the patients were scheduled for a computed tomography (CT, Aquilion, Toshiba Corp., Tokyo, Japan) (120 kVp; 350 mA; rotation time, 0.5 s; slice thickness, 0.5 mm) scan 3 weeks before the surgery. The image software packages Rhinoceros 5.0 (Robert McNeel & Associates, Seattle, WA, USA) and Geomagic Studio (12th edition; Geomagic, Inc., Cary, NC, USA) were used for the image processing and the setup of the orientation systems. The jawbone repositioning movements were based on the maxillomandibular complex (MMC), with stable occlusion set by orthodontists. The overall feasibility of the simulated plans was finally approved by the same surgeon (J.P. Lai).

### 2.2. The Guidance System and the Intraoperative Procedures

The simulated results were carried out intraoperatively with the help of the guidance system, which consisted of one pair of guidance plates (Figure 1a,b), one occlusion registration stent, and the comprehensive prebent fixation plates (Figure 2). The CAD/CAM technique was used to help compliance with the virtual planning. One pair of guidance plates carrying the information about the cutting orientation and thickness were virtually designed and then printed out using biocompatible resin. Subsequently, the fixation plates over bilateral medial and lateral maxillary buttress were prebent, and served to guide the detailed maxillary repositioning according to the screw holes of the guidance plates. At last, the occlusion registration stent was used to settle the planned mandibular orientation and position, which were then fixed by the fixation plates.

### 2.3. The Protocol of Postsurgical Care and Subsequent Clinical Follow-Up

After the surgery, the patients received an identical postsurgical care protocol, consisting of 2 to 4 weeks of intermaxillary fixation (IMF), and semi-rigid IMF by elastics for additional 2 to 4 weeks after removing the occlusion registration stent. Afterwards, the patients were instructed in mouth-opening practice to recover their normal mouth-opening range. Most of the patients initiated the postsurgical orthodontic adjustments in the second or third month after the surgery. In the sixth month after the surgery, all of the patients received another craniofacial CT for the outcome assessment.

### 2.4. The Cephalometric Check-Up for the Immediate Postsurgical Accuracy

During the first week after the surgery, the patients received a cephalometric assessment to check the accuracy of jawbone repositioning. For comparison with the measurements form the virtual planning, the patients’ heads were carefully oriented before taking the cephalometric films. The orientation principles included paralleling the Frankfort plane with the ground and adjusting the head position by centering the line passing through glabella and the mid-interpupil point. The distance from the A-point to the nasion perpendicular line (A–Nv) verified the repositioning error in the sagittal direction, and the distance between the A-point and the Frankfort plane (A–FHP) indicated the discrepancies in the vertical axis. Both measurements were performed using the AudaxCeph Empower software (Version 5.2, Ljubljana, Slovenia). Both cephalometric measurements were used to verify the possible repositioning errors by comparison with the measurements from presurgical virtual planning. Cases presenting > 2 mm differences either sagittally or vertically were regarded as modified intraoperative procedures, and were excluded from the study.

### 2.5. Verifying the Jawbone Changes by Using the Representative Triangles

In the present study, a representative triangle amid the virtual maxillary anatomical boundaries was depicted in the simulation stage. The A-point was at the top of the triangle. On the same transverse plane, the tangent lines passing through the A-point were used to identify the most lateral points, the maxillary right border (MxR), and the maxillary left border (MxL). The same representative triangle could be duplicated at different stages by transferring the three-dimensional information with superimposition over the area of the posterior nasal spine (PNS). Thereafter, the linear and angular changes of the landmarks or reference planes at different stages could be measured (Figure 3) [8].

### 2.6. The Reliability of Virtual Simulation in the Transverse Dimension

In present study, the planned surgical movements (PSMs) and the outcome discrepancy (OD) were measured to verify the predictability of jawbone changes in the transverse dimension. Taking the representative triangle of the simulation stage as the baseline, the location and orientation differences of each landmark and the reference lines on the representative triangles at different stages were measured for verification. The PSM was retrieved by overlapping the representative triangle of the original stage onto the one of the simulation stage; on the other hand, the overlapping differences between the actual outcome of the craniofacial CT taken at the sixth month after surgery and the simulation stage revealed the OD. The linear and angular changes of the A-point were automatically calculated by the software to avoid manual measurement errors (Figure 3).

### 2.7. The Statistical Analysis of the Present Study

In order to test the research aims of the present study, Spearman’s correlation was used to verify the correlation between PSM and OD. The sample power and the normality distribution were tested using the G*Power software (version 1.1, University of Düsseldorf, Düsseldorf, Germany) and Kolmogorov–Smirnov analysis. Additionally, the intra- and inter-examiner measurement reliability was also validated by randomly selecting 10 samples to repeat the proposed measurements, at a time interval of 2 weeks. The intraclass correlation coefficients were adopted for the reliability validation.

## 3. Results

In total, 44 adult patients were enrolled into the study. The average age of the patients was 21.95 ± 4.1 years old. The average PSM of the A-point in the transverse axis was 1.04 ± 0.92 mm, and the average yaw correction was 1.44° ± 1.23°. Meanwhile, the average OD of the A-point was 0.66 ± 0.54 mm, and the overall yaw correction showed a difference of 1.01° ± 0.84° (Table 1). A post hoc power analysis revealed that all of the measurements possessed strong power in sample size (α< 0.05, one-tailed at a power of greater than 0.99).

The immediate post-OGS cephalometric measurements showed differences between the planned and postoperative positions in both the sagittal and vertical axes. The average difference of A–Nv was 0.38 ± 0.15 mm, while that of A–FHP was 0.72 ± 0.48 mm.

The normality test proved the heterogeneity among the samples. Thus, non-parametric tests were used. The results from Spearman’s correlation showed non-specific relevance between all of the linear PSMs and ODs. However, there was a positive correlation in the angular correction; that is, with planning and performance of yaw correction, more ODs were observed (Table 2).

Finally, the intra-examiner and inter-examiner reliabilities were in agreement (0.978 and 0.972, respectively) with the error range of 0.00–0.06 mm (mean: 0.023 ± 0.007 mm).

## 4. Discussion

This study aimed to investigate the OGS outcome predictability in the transverse dimension. All of the enrolled subjects possessed a Class III skeletal pattern. With sufficient sample power, we compared the postsurgical results with the simulated plans to determine the ODs. According to the results, most subjects (43/44) revealed a linear difference of <2 mm—known as the threshold of clinical relevance—6 months after the OGS [9]. Clinically, such a finding implies that the surgical outcome in the transverse direction is highly predictable according to the virtual planning. In consideration of the fact that major postsurgical relapse usually takes place 3 to 6 months after surgery [4,10,11], we took the sixth-month postsurgical CT to evaluate the outcomes observed clinically. The clinical ODs could be attributed to the intraoperative repositioning errors and the postsurgical instabilities of the jawbones.

### 4.1. Intraoperative Repositioning Errors

In this study, the results of virtual planning were transferred via the waferless technique [3,4,5,6]. Previous reports using the waferless technique have reported promising accuracy of jawbone repositioning. With the help of intraoperative surgical navigation, Zinser et al. reported less than 0.35 mm error in sagittal movements and less than 0.64 mm error in vertical ones [6]. Subsequently, Kim et al. reported 0.03 mm and 0.4 mm discrepancy in the sagittal and vertical axes, respectively, according to the immediate postsurgical CBCT results [7]. Based on these informative results, in our study, we chose the post-OGS cephalometric measurements as the validation tool, without additional radiation exposure. The cephalometric results revealed comparable intraoperative repositioning errors in both the sagittal and vertical axes. Therefore, it would be reasonable to infer that the errors in the transverse axis would be similar to the reported data. The jawbone repositioning in the transverse axis was reported to be highly accurate, with minimal error, ranging from 0.07 mm [12] to 0.3 mm [7].

### 4.2. Postsurgical Instabilities

No matter how the hardware evolves, the biological resistance that triggers postsurgical instabilities or relapse is always the challenge to be faced. Postsurgical changes have been widely discussed in the literature [13,14,15]. Several possible confounding factors have been discussed, although lateral cephalometric films have been used for measurements. For example, the skeletal pattern [13], one- or two-jaw surgery mostly [16,17,18], the surgical methods [19,20,21], the fixation methods [22,23], and the combinations of jawbone movements are reportedly related to the postsurgical changes. Therefore, we collected samples of the same characteristics receiving identical surgical protocols to achieve improved facial appearance. In addition, another study conducted by our team presented stable postsurgical temporomandibular joint positioning in patients with a Class III skeletal pattern by using the same computer-assisted simulation system [24]. Under such circumstances, we found that the simulated results from virtual planning could be reliably predicted and fulfilled in the transverse dimension in the surgical correction of Class III facial pattern.

In addition, other confounding factors were also worthy of discussion. In previous studies, the positive relationship between the linear PSM and postsurgical changes in the vertical and sagittal directions was reported [13]. That is, more linear surgical movements carried out intraoperatively translated into more vulnerable postsurgical stability. Such a phenomenon could be explained by the decreased resistance of the fixation plate against soft tissue traction during the healing period [25]. However, in our study, such a correlation did not exist in the transverse dimension (Table 2). This could be explained by the availability of a wider range of bony support and fixation in the transversal correction.

For the angular correction, according to the results, the average OD did not reach clinical significance, reported at 4° in the literature [26]. However, the positive correlation between the PSM and OD in yaw correction implied higher discrepancy resulting from greater planned movements. Therefore, it is suggested that practitioners should coincide the dental midline with the facial midline as precisely as possible in the presurgical preparation, instead of taking additional yaw correction just for the purpose of midline coincidence. Such a finding echoes a previous study suggesting the elimination of unnecessary yaw movements in the orthognathic treatments of facial prognathism—especially for patients with facial asymmetry and craniofacial anomaly [27].

There are certain limitations of this study. First, our results only focused on the anterior maxillary movement. However, the strain of pterygomandibular sling might make the posterior maxilla vulnerable to outcome differences. The posterior maxillary movement should be investigated in future surveys. Second, the mandibular ODs were not discussed in the study. In modern virtual planning systems, the planned jawbone positioning is commonly achieved with simultaneous adjustments of the MMC. The limited change in the maxilla might still cause remarkable differences in the mandible. Finally, only patients treated with one-piece LeFort I surgery were enrolled in the study. However, the situation might be different when patients are treated with segmented LeFort I osteotomy, which has been regarded as a highly unstable procedure.

## 5. Conclusions

With the help of presurgical virtual planning, practitioners can predict the surgical outcomes of linear changes in the transverse direction in the OGS of Class III patients. Practitioners should be advised to avoid unnecessary yaw correction for more predictable postsurgical results.

## Figures and Tables

**Figure 1 jpm-12-01147-f001:**
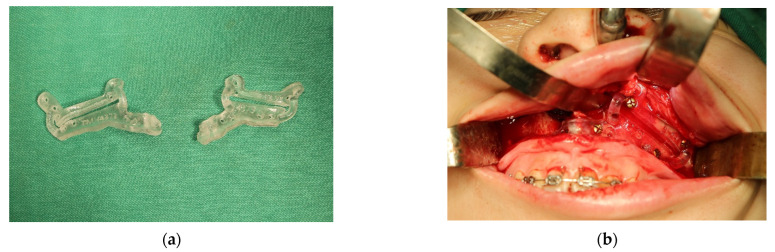
(**a**) The 3D-printed surgical guides recorded the orientation and thickness of the surgical cuts. The screw holes over the surgical guides also served to guide the placement of the prebent fixation plates after jawbone repositioning. (**b**) The guidance plates were accurately affixed onto the maxillary surface to provide cutting guidance.

**Figure 2 jpm-12-01147-f002:**
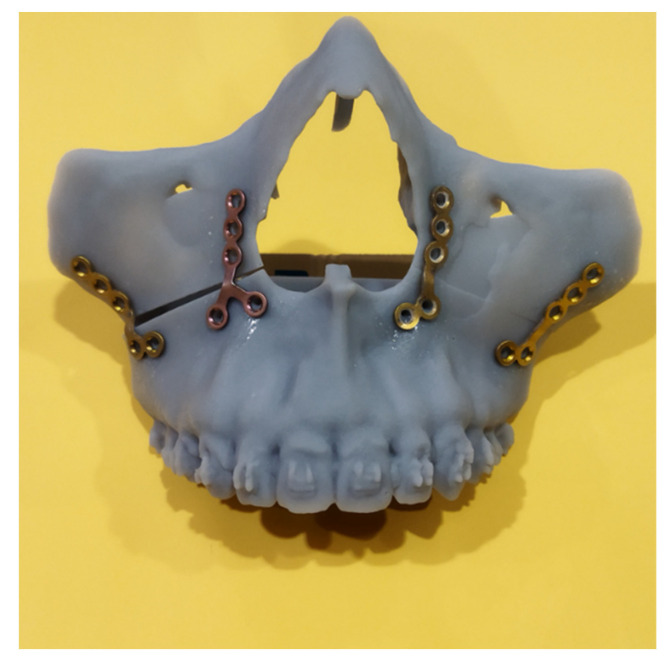
The intraoperative fixation plates were prebent on a 3D-printed stereolithographic model of actual anatomical size. The precise matching of the screw holes and fixation plates provided guidance for the intraoperative jawbone repositioning and orientation.

**Figure 3 jpm-12-01147-f003:**
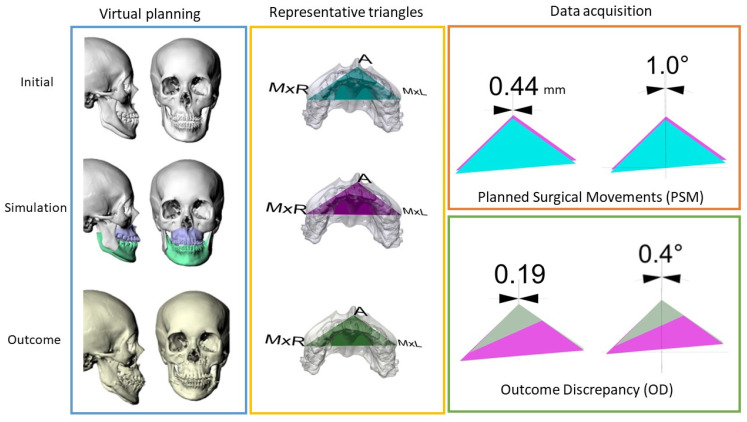
The illustration of the processing procedures, including virtual planning and outcome assessments: A representative triangle was incorporated into the maxilla and transferred into different stages by registering it at the area of the posterior nasal spine. The representative triangle at the simulation stage (pink) served as the reference plane. The planned surgical movements were calculated by overlapping the initial triangle (blue) onto the reference plane. On the other hand, the outcome discrepancy was measured by the differences between the outcome triangle (green) and the simulation triangle.

**Table 1 jpm-12-01147-t001:** Translational movement of the A-point and angular change of the maxilla in the transverse axis.

		Transverse (mm)	Yaw (^°^)
Planned surgical movement	Mean	1.04 ± 0.92	1.44 ± 1.23
	Min	0.00	0.00
	Max	3.88	4.13
Outcome discrepancy	Mean	0.66 ± 0.54	1.02 ± 0.84
	Min	0.00	0.08
	Max	2.20	3.73

**Table 2 jpm-12-01147-t002:** Spearman’s correlation of PSM and OD in transverse and yaw: Spearman’s correlation revealed that the outcome discrepancy in yaw rotation had a positive relationship with planned surgical movement in yaw rotation (PSM: planned surgical movement; OD: outcome discrepancy; *p* = 0.05; *: statistically significant).

		PSM-Transverse	PSM-Sagittal	PSM-Vertical	PSM-Roll	PSM-Pitch	PSM-Yaw
OD-transverse	Correlation coefficient	0.158	0.033	−0.195	0.099	−0.040	0.021
	Significance	0.307	0.834	0.205	0.522	0.796	0.890
OD-yaw	Correlation coefficient	0.088	−0.289	0.062	0.029	0.026	0.315
	Significance	0.569	0.058	0.689	0.851	0.864	0.037 *

## Data Availability

The data presented in this study are available upon request from the corresponding author. The data are not publicly available, due to the privacy protection of the studied individuals.

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
