# Peer review of "The Predictability of the Surgical Outcomes of Class III Patients in the Transverse Dimension—A Study of Three-Dimensional Assessment"

_jpm, 2022, doi:10.3390/jpm12071147_

Round 1

Reviewer 1 Report

Dear authors, 

In the introduction section you mention the waferless technique. Did you use it? A more detailed description of the used thechique should be added. 

You mention the postsurgical changes. Did you follow the relapse? Or the change in the position of the condyle? It would have great clinical impact if you could mention it. 

How did you etablish the predictability of planned maxillary surgical movements? 

Which are the confounding factors that affect the predictability of surgical outcomes in the transverse dimension?

In the abstract it is not necessary to mention all 200 patients, the study sample would be enough. 

Please add some pre-surgical virtual planning images. 

Also, add details regarding cephalometric check-up for immediate postsurgical accuracy. 

The representative triangles should be also presented as images.

Some details regarding patient follow up should be added.  

Reviewer 2 Report

The article concerns an interesting and current topic of virtual orthognathic surgery planning. It is extremely important to determine the possibility of transferring the planned movements of the jaw bones and their correlation with the real treatment outcomes achieved. Depending on the navigation system used, the outcome discrepency may vary.

The abbreviation OGS is introduced for the first time in the abstract and has no full form in front of this (line 25).

The article would be more interesting if the authors attached figures illustrating the method of assessing outcome discrepencices through the superimposition of images before and after surgery. The very description of this methodology should also be made in more detail. (lines 123-131)

The authors should also discuss the following issues:

1. Why were only patients with class III skeletal pattern eligible for the study, some class II defects are also treated by bimaxillary approach with clockwise or counterclockwise rotation of the maxillomandibular complex?

2. Were class III patients with asymmetries in the maxilla or mandible excluded from the study? The asymmetries could cause quite significant influence on the outcome discrepancies.

3. Were the surgeries carried out in the maxilla first or mandible first manner. This could greatly influence the outcome discrepancies.

Round 2

Reviewer 1 Report

Dear authors, 

The manuscript has been significantly improved.

In line 139 maybe you wanted to say - aimed

It would be interesting to report the data as well 1 year after surgery and to compare the results.

Good luck with your work!

Reviewer 2 Report

The authors have answered all my concerns about the original version of the manuscript, so that I am satisfied with the current version of the paper.